# Stabilization Mechanism of Semi-Solid Film Simulating the Cell Wall during Fabrication of Aluminum Foam

**Takashi Kuwahara [1],\*, Akira Kaya [1], Taro Osaka [1], Satomi Takamatsu [1] and Shinsuke Suzuki [1,2]**

[1] Faculty of Science and Engineering Faculty of Science and Engineering, Waseda University, 3-4-1 Okubo, Shinjuku, Tokyo 169-8555, Japan; s49y5wrgw-sl@suou.waseda.jp (A.K.); t-evolva@moegi.waseda.jp (T.O.); s-takamatsu@asagi.waseda.jp (S.T.); suzuki-s@waseda.jp (S.S.)

[2] Kagami Memorial Research Institute of Materials Science and Technology, Waseda University, 2-8-26 Nishi-Waseda, Shinjuku, Tokyo 169-0051, Japan

\* Correspondence: takuwahara@ruri.waseda.jp; Tel.: +81-3-5286-8126

**Abstract:** Semi-solid route is a fabrication method of aluminum foam where the melt is thickened by primary crystals. In this study, semi-solid aluminum alloy films were made to observe and evaluate the stabilization mechanism of cell walls in Semi-solid route. Each film was held at different solid fractions and holding times. In lower solid fractions, as the holding time increases, the remaining melt in the films lessens and this could be explained by Poiseuille flow. However, the decreasing tendency of the remaining melt in the films lessens as the solid fraction increases. Moreover, when the solid fraction is high, decreasing tendency was not observed. These are because at a certain moment, clogging of primary crystals occurs under the thinnest part of the film and drainage is largely suppressed. Moreover, clogging is occurring in solid fraction of 20–45% under the thinnest part of the film. Moreover, the time to occur clogging becomes earlier as the solid fraction increases.

**Keywords:** porous metal; semi-solid; aluminum foam; primary crystals; drainage; clogging

---

## 1. Introduction

Recent days, improved fuel consumption is required for transportation equipment such as automobiles. Furthermore, passenger safety including the crash safety must be maintained. For this purpose, aluminum foams with many pores dispersed inside are actively investigated. They are ultralight materials and exhibit shock absorbing characteristics called plateau phenomenon when compressed. Additionally, aluminum foam can absorb shocks isotropically [1]. Therefore, it is expected to be applied for transportation equipment [2]. Moreover, they have excellent property in sound insulation equivalent to glass wool [3,4]. Furthermore, they also have heat insulation properties [5]. Furthermore, it can be applied for architecture material [6].

However, there is a problem of drainage during fabricating aluminum foam which coarsens pores and thus, deteriorates the mechanical properties. Usually, aluminum foams are obtained by solidifying the aluminum alloy melt with pores generated inside. During holding of the foam before solidification, liquid flows downwards in a cell wall (a film between pores) due to the gravity. This phenomenon is called drainage. Due to the remarkable collapse of cell walls, pore morphology becomes uneven. Moreover, unfoamed parts would appear at lower part of the foam. However, by suppressing the drainage by thickening the melt, foam with uniformed pore morphology could be obtained [7]. Fabrication of foams with uniform pores becomes possible by stabilizing the cell walls through thickening.

There are several routes to fabricate aluminum foams. Melt-route, shown in Figure 1 is the most known route to fabricate aluminum foam. First, the melt is thickened by oxide particles. Then, the melt is stirred with adding blowing agent ($TiH_2$) included. Finally, an aluminum foam can be obtained by foaming and water cooling. However, our group is fabricating foams in the Semi-solid route shown in Figure 1. The fabrication method is almost the same as the Melt route except for thickening the melt by primary crystals existing in semi-solid state [8]. In the Semi-solid route, the adding of thickener, such as ceramic particles and Ca, can be excluded. Therefore, foam with less impurities can be obtained. Moreover, foams obtained by the Semi-solid route are thought to have higher compressive yield stress than that obtained by the Melt route [9].

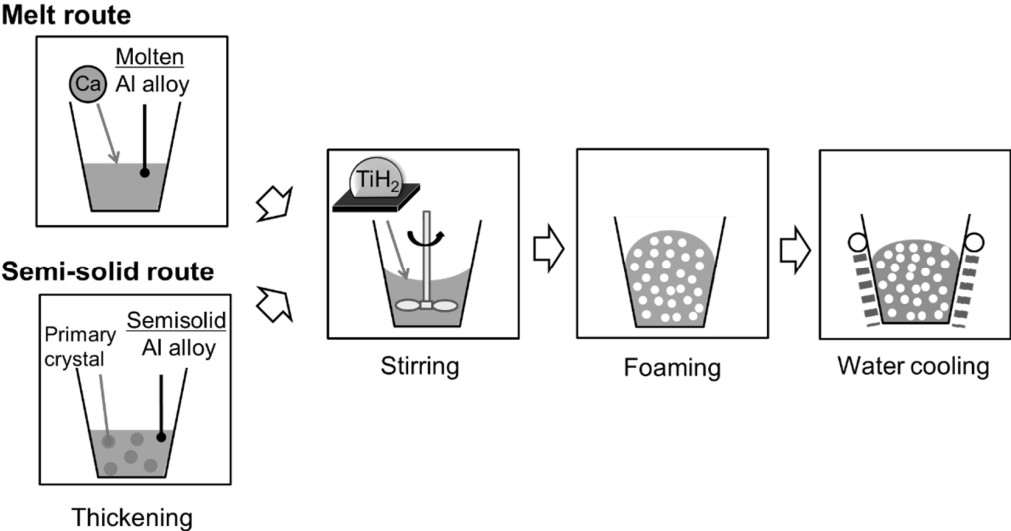

**Figure 1.** Fabrication methods of aluminum foam (Melt route and Semi-solid route).

Jin et al. evaluated the Melt route with SiC particles as a thickener and elucidated that the parameters to fabricate stabilized foams (foams with stabilized cell walls) are particle size and particle fraction. Similarly, they expressed the parameter map for fabricating of stabilized foams [10]. Moreover, Heim et al. fabricated foams with various types of thickener including primary crystal and classified them as foamable, partially foamable, or unfoamable. They also expressed that the parameter map proposed by Jin et al. could be adapted to various types of thickener [11].

Moreover, in the Melt route, Heim et al. expressed that some thickener particles are covered by oxide skin of cell wall and interact and builds bridge with other particles resulting to prevent drainage [12]. This is thought to be the stabilization mechanism of cell wall in the Melt route. However, the stabilization mechanism of cell wall in the Semi-solid route has not been evaluated. Furthermore, size of primary crystals is quite large and out of the stabilization region in the parameter map proposed by Jin et al. [10]. Likewise, primary crystals were classified as partially foamable or unfoamable by Heim et al. [11]. Despite, our group found it possible to fabricate stabilized foams in the Semi-solid route out of the stabilization region in the map. Therefore, cell wall of Semi-solid route is thought to have different stabilization mechanism from Melt route. Therefore, elucidation of stabilization mechanism of cell wall of Semi-solid route is required.

However, it is difficult to evaluate the cell walls since the size and shape of cell walls are uneven. As a solution, Heim et al. simulated the ideal shape of the cell wall using aluminum alloy film [13]. In this method, we can evaluate the cell wall in higher comparability. Therefore, the objective of this study is to elucidate the stabilization mechanism of cell wall of the Semi-solid route using aluminum alloy film.

## 2. Materials and Methods

### 2.1. Sample Material

Sample material used in this study was Al-6.4mass%Si. Since the phase diagram of Al-Si is simple and primary crystals of hypoeutectic alloy are easy to observe, this material is suitable for evaluation of stabilization mechanism. The Al-6.4mass%Si ingots were prepared by mixing pure aluminum with Al-25mass%Si. The compositions of the pure aluminum and the Al-25mass%Si ingots used in this study are shown in Tables 1 and 2, respectively.

**Table 1.** Composition of pure aluminum.

| Element | Si | Fe | Cu | Al |
|---|---|---|---|---|
| mass% | 0.004 | 0.003 | 0.001 | bal. |

**Table 2.** Composition of Al-25mass%Si.

| Element | Si | Fe | Cu | Al |
|---|---|---|---|---|
| mass% | 25.1 | 0.17 | 0.00 | bal. |

### 2.2. Forming of Aluminum Alloy Film

Aluminum alloy films which are simulating the cell walls of aluminum alloy foam were formed to evaluate the stabilization mechanism of cell wall in higher comparability. In the Semi-solid route, the melt was thickened in semi-solid state as shown in Figure 1. Therefore, aluminum alloy films were also formed in the semi-solid state to observe the cell walls in the Semi-solid route. The solid fractions of the melt of Al-6.4mass%Si were obtained accurately using Thermo-Calc 2019a (Itochu Techno-Solutions Corporation, Tokyo, Japan) as shown in Figure 2.

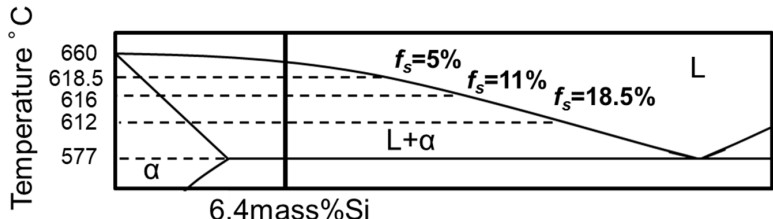

**Figure 2.** Al-Si binary phase diagram. The values of solid fraction were calculated by Thermo-Calc.

All of processes to form aluminum alloy films were done inside a chamber under pressure condition of 5000 Pa of air to suppress heat dissipation from the film during the retaining process described later. To obtain this condition, pressure was adjusted by vacuuming continuously while controlling the leaking rate with valve. To form the aluminum alloy film, Al-6.4mass%Si was completely molten in the crucible. Then, the melt was slowly cooled to the semi-solid temperature. The temperature was measured by K-type thermocouple coated with heat resistant inorganic adhesive (Toagosei Co. Ltd., Tokyo, Japan) soaked in the melt. Then, the melt was stirred for 60 s at 900 rpm so as to round the shape of primary crystals [14]. Before this process, thermocouple was taken out of melt to make the stirring possible. Since the stirring generates the temperature of melt to change, the thermocouple was soaked again in the melt and the temperature was adjusted again to semi-solid temperature.

After the semi-solid temperature reached the setting temperature, a wire shown in Figure 3a was dipped into the melt. This wire was shaped so as to form a liquid film between the rings when pulled up from the melt. The design of this wire originates from a previous research [13]. In this study, it was adopted to study the semi-solid densification phenomena. The material of the wire was stainless steel SUS304 and the drainage channel was provided for the melt to flow down from the film and

reproduce the drainage. The ratio of distance between the rings versus its diameter was one versus three following the previous study to form a film with a continuous curvature [13]. After pulling up the film from the melt, the film was retained over the melt with drainage channels contacted with the melt to reproduce the drainage [13]. This process simulates the progression of drainage which causes collapse of cell walls in aluminum form. The holding time was changed in each film. Finally, the aluminum alloy film was obtained by pulling up the film at 5 mm/s and rapid cooling by dipping into a melt pool of low melting point alloy U78 (Osaka Asahi Co. Ltd., Osaka, Japan) held at 150–200 °C in the chamber. The schematic of the experimental procedure is shown in Figure 4.

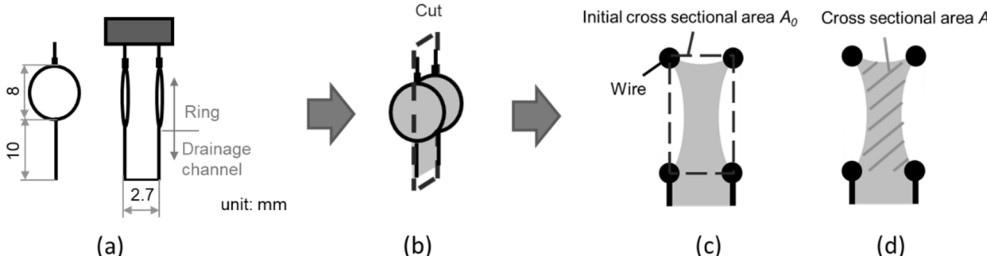

**Figure 3.** Schematic of wire and film: (**a**) size of wire; (**b**) cutting of film; (**c**) measurement of initial cross-sectional area; and (**d**) measurement of cross-sectional area.

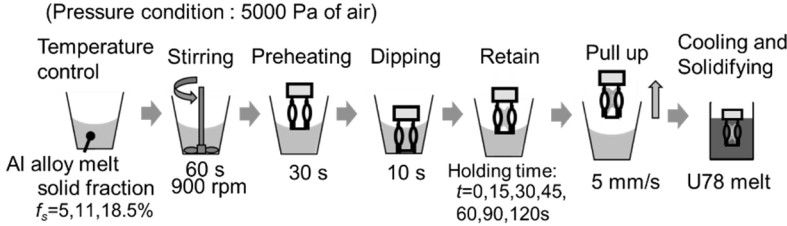

**Figure 4.** Schematic of forming liquid film of aluminum alloy.

Holding times of the aluminum alloy films were 0, 30, 60, 90, and 120 s. However, since it took a little time to move film to cool down, actual time of drainage was longer than the time designated. The solid fractions of melt were 5%, 11%, and 18.5%. Aluminum alloy film was formed for each combination of holding time and solid fraction. For solid fraction of 5%, holding time of 15 s and 45 s were also obtained for further consideration.

### 2.3. Observation of Aluminum Alloy Film

The formed film was first rinsed inside with the boiled water to remove U78 used for rapid cooling. Then, formed films were cut in the middle as shown in Figure 3b and the cross sections were observed by optical microscope. Primary crystals with a diameter larger than 200 μm, which seemed to have existed in a semi-solid state, were colored on micrographs with using an image processing software GIMP 2.10.4 (The GIMP Development Team, Berkeley, CA, USA). Then, the cross-sectional area of the films and the area of colored primary crystals were measured with using an image processing software WinROOF™ 6.4.0 (Mitani corporation, Fukui, Japan).

The cross-sectional area was measured to evaluate the change of melt quantity during the holding time. Furthermore, comparison of cross-sectional area was done by normalized cross-sectional area, as a parameter of the area of the remained melt compared with the initial one. To obtain the normalized cross-sectional area, initial cross-sectional area $A_0$ is defined as area surrounded with wires (Figure 3c). Then the normalized cross-sectional area was obtained through dividing the cross-sectional area A (Figure 3d) by the initial cross-sectional area. However, small deformation occurred on wires because of the flow stress of the semi-solid state. Therefore, the wires of the actual results were not aligned as schematic in Figure 3a. Likewise, this tendency is remarkable in the lengths between the rings while the lengths from the diameter of rings show relatively close value. Therefore, the initial cross-sectional

area was obtained by calculating the area of trapezoid. The upper and lower length between the rings were used as upper and lower bases of the trapezoid. The average diameter of rings was used as its height.

## 3. Results

### 3.1. Temporal Change of Cross Sectional Area

The plots in Figure 5 show the temporal change of the normalized cross-sectional area obtained from the experimental results. From Figure 5, the normalized cross-sectional area decreases as the holding time increases at lower solid fractions. Additionally, decreasing tendency of the normalized cross-sectional area lessens as the solid fraction increases. Moreover, at high solid fraction this decreasing tendency was not observed.

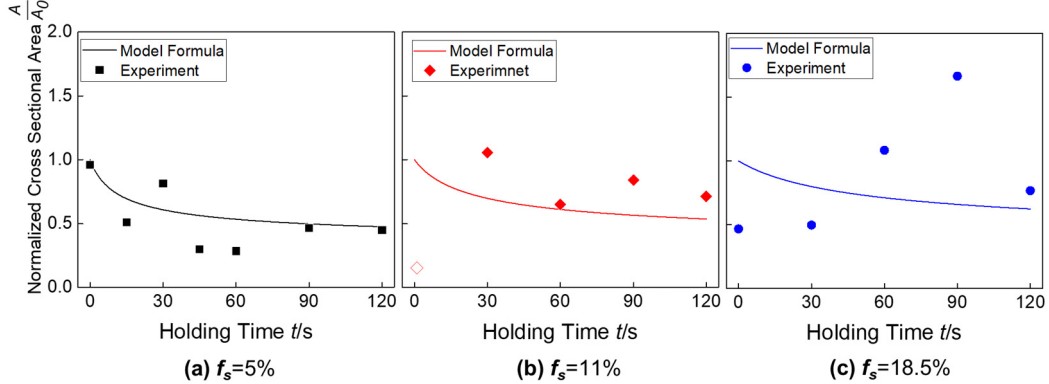

**Figure 5.** Temporal change of normalized cross-sectional area: (**a**) solid fraction $f_s$ = 5%; (**b**) $f_s$ = 11%; and (**c**) $f_s$ = 18.5%.

### 3.2. Observation of Primary Crystals

Figure 6b shows the cross-sectional photomicrograph of an aluminum alloy film shown in Figure 6a. In Figure 6c, primary crystals that seem to have been existing in the semi-solid state are colored as expressed in Section 2.3. Most of them are observed in the lower part of the film in the micrograph and less primary crystals were observed in the drainage channel than in the ring. Therefore, primary crystals seem to have clogged in the lower part of the film. However, only for the film for $f_s$ = 11%, $t$ = 0 s, primary crystals that seem to have been existing from semi-solid state were not observed (Figure 7). This is an exception. In this experiment, the part with a small solid fraction might have been pulled up for some reason, such as inhomogeneous solid fraction in the crucible.

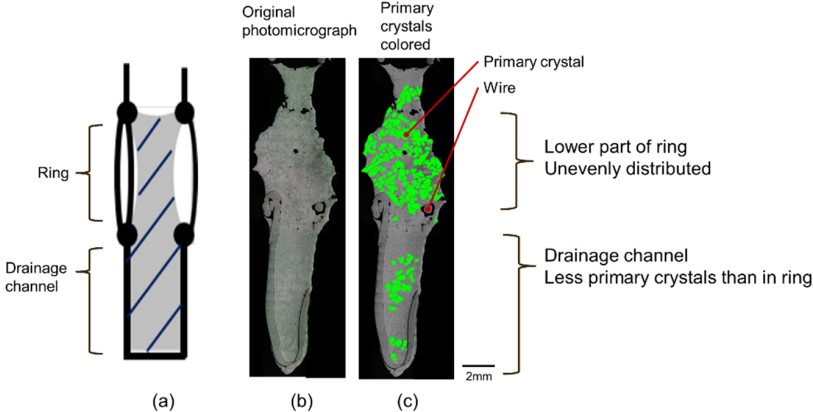

**Figure 6.** Cross section of aluminum alloy film (solid fraction $f_s$ = 18.5%, holding time $t$ = 120 s): (**a**) cross section in film; (**b**) photomicrograph; and (**c**) primary crystals colored on photomicrograph.

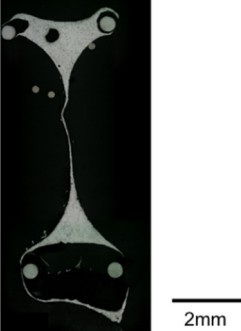

2mm

**Figure 7.** Photomicrograph of cross section of aluminum alloy film (solid fraction $f_s$ = 11%, holding time $t$ = 0 s).

## 4. Discussion

### *4.1. Suppression of Drainage by Thickening*

As described in Introduction, thickening of melt stabilizes the cell walls and could homogenize the pore morphology. Increase of viscosity by thickening suppresses the liquid flow in aluminum alloy film. This is thought to be the stabilization mechanism of cell wall. For confirmation, we derived a model formula of melt flow and compared it with the experimental results. The model formula describes relationship between the holding time and the normalized cross-sectional area of the film.

#### 4.1.1. Derivation of Model Formula

Figure 8 shows the change in geometry of the aluminum alloy film. Heim et. al. approximated the cross-sectional shape of the pulled-up film as a rectangle [12]. Then, the film will start to curve as the drainage progresses. The curvature radius of the film is decided by approximation to an arc defined by three points; the top and bottom of the wire and the middle of the film. The middle point of the film should shrink as the drainage progresses. Therefore, the curvature radius of the film should change continuously. The depth of the film $L$ would not change in this model. This is because from observation of films including Figure 6, change in depth of film is small compared to width of film $x_0$. This may be because it is difficult to make curvature radius smaller when the original curvature radius is small. In this situation, the relationship between the cross-sectional area $A$ of film and width of film $x_t$ is shown in Equation (1). (The discussion is shown in Appendix A).

$$A = hx_0 - \frac{1}{4\Delta x}\left\{\frac{\pi}{2\Delta x}\left(h^2 + \Delta x^2\right)^2 \frac{\theta}{360°} - \left(h^2 + \Delta x^2\right)h\right\} \tag{1}$$

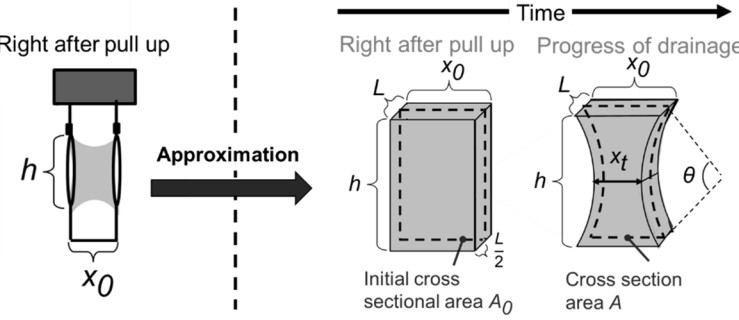

**Figure 8.** Geometry model of aluminum alloy film.

Here, $\theta$ is curvature radius of the film, $h$ is height of rectangle, $x_0$ is width of rectangle, and $x_t$ is width of middle point of rectangle which would change continuously. Furthermore, $\Delta x$ is defined as Equation (2).

$$\Delta x = x_0 - x_t \tag{2}$$

Moreover, the curvature radius of the film $\theta$ can be shown in Equation (3).

$$\theta = \mathrm{Arcsin}\frac{4h(x_0 - x_t)\left\{h^2 - (x_0 - x_t)^2\right\}}{\left\{h^2 + (x_0 - x_t)^2\right\}^2} \tag{3}$$

However, since the changing parameter is width of film at middle $x_t$, Equation (1) could not explain the temporal change of the cross-sectional area of film $A$. Therefore, a fluid dynamic model of film is considered. Brady et. al. considered that liquid (semi-solid) flow in a film by drainage can be described as the Poiseuille flow under condition that the ends of both sides are fixed [15]. The situations are, one way flow, steady flow, and gravity as the only external force, as shown in Figure 9a. Following the Poiseuille flow, width of film at middle $x_t$ can be shown as Equation (4).

$$x_t = x_0\sqrt{\frac{6\mu h}{\rho g t x_0^2 + 6\mu h}} \tag{4}$$

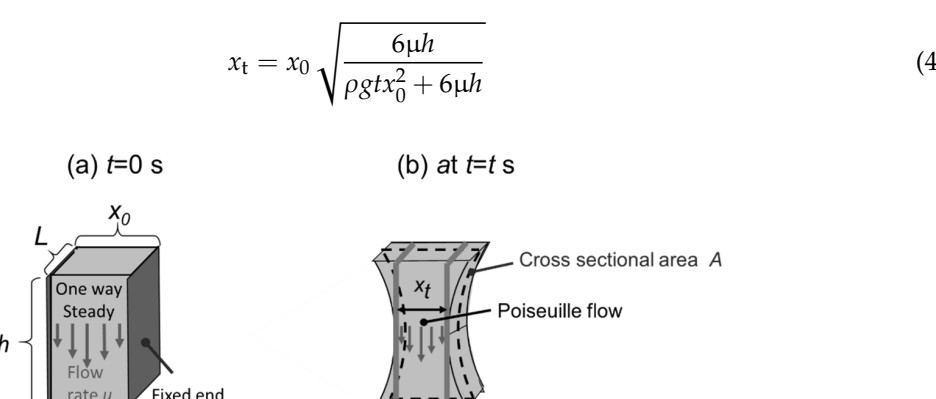

**Figure 9.** Schematics of Poiseuille flow and deformation of the film: (**a**) at $t = 0$ s and (**b**) at $t$.

Here, $\mu$ is the viscosity of melt, $\rho$ is density of melt and $g$ is the gravitational acceleration. By Equation (4), relationship between the width of film at middle $x_t$ and holding time $t$ could be obtained. By substituting the width of film at middle $x_t$ obtained in Equation (4) into Equation (2), relationship between the cross-sectional area of film $A$ and holding time $t$ could be obtained by Equations (1) and (3). Therefore, a model formula to compare with the result is obtained. Figure 9b shows a schematic model of the phenomenon expressed by this model formula. Correspondingly, in this model, since the mainly considered area is middle of the film, which is assumed that drainage occurs mainly, three-dimensional deformation and drainage from ends of film is not considered.

### 4.1.2. Comparing Cross Sectional Areas Obtained from the Model with Experimental Results

To compare the model and the experimental results, the model formula was transformed into a dimensionless formula shown in Equation (5). $A_0$ in Equation (5) is the initial cross-sectional area and can be shown in Equation (6).

$$\frac{A}{A_0} = 1 - \frac{1}{4hx_0\Delta x}\left\{\frac{\pi}{2\Delta x}\left(h^2 + \Delta x^2\right)^2\frac{\theta}{360°} - \left(h^2 + \Delta x^2\right)h\right\} \tag{5}$$

$$A_0 = hx_0 \tag{6}$$

Figure 5 shows the cross-sectional areas obtained from the model with experimental results. According to the experimental data of Al-6.5mass%Si obtained by Moon et. al. using a Searle-type viscometer [16], the viscosity values of 22, 45, and 100 mPa·s were used for 5%, 11%, and 18.5% of solid

fraction, respectively. Furthermore, the density value used was $2.4 \times 10^3$ kg/m$^3$ for every solid fraction. This value was assumed from study done by Kudoh et. al. which measured the density of liquid phase in semi-solid state of Al-2.4mass%Si [17]. However, change of density between $2.3$–$2.7 \times 10^3$ kg/m$^3$ only produces about 3% change in the calculated value of the model formula and has no effect on discussion. At lower solid fractions, experimental results seem to fit model formula. However, fitting becomes worse as the solid fraction increases. Furthermore, for high solid fraction, fitting of model formula to experimental results was not good. Therefore, it is difficult to conclude that only the increase in fluid viscosity is the mechanism to stabilize the film. Therefore, consideration of other factors to suppress the drainage is required.

## 4.2. Suppression of Drainage by Clogging of Primary Crystals

As expressed in Section 3.2, primary crystals seem to have been clogging in the lower part of the film. To confirm and to find the clogging part more finely, the distribution of primary crystals in the films was measured in the height direction from the lower wires (Figure 10). In Figure 10, the primary crystals that seem to have been existing from semi-solid state are colored as expressed in Section 2.3. As the primary crystals were found more in the lower part than in the upper part and drainage channel, clogging should have occurred in the lower part under the thinnest part of the film. Thus, this factor effects to suppress drainage more strongly than thickening. Likewise, at solid fraction of 5%, this tendency was seen remarkably in films with long holding time.

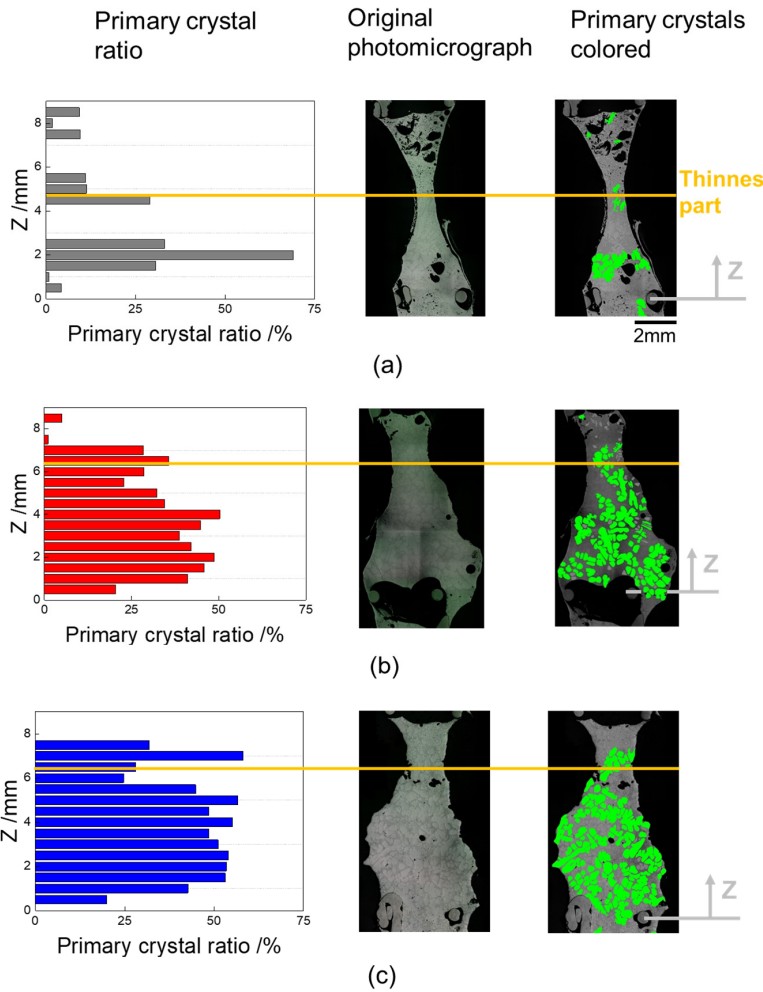

**Figure 10.** Primary crystal ratio: (**a**) solid fraction $f_s$ = 5%, holding time $t$ = 120 s; (**b**) $f_s$ = 11%, $t$ = 90 s; and (**c**) $f_s$ = 18.5%, $t$ = 120 s.

To derive this clogging mechanism more quantitatively, the critical solid fraction $f_{s\_cr}$ for clogging was evaluated. Areas of primary crystals $A_p$ and film $A_f$ were measured in the region between the thinnest part of the film and the lower wires (Figure 11). In Figure 11, primary crystals that seem to have been existing in semi-solid state are colored in the same way as expressed in Section 2.3. The area of aluminum alloy film is also colored in the same way as expressed in Section 2.3. By dividing the area of primary crystals $A_p$ by area of aluminum alloy film $A_f$, the critical solid fraction for clogging was obtained. Figure 12 shows the measured results of the critical solid fraction for clogging. In Figure 12, the critical solid fraction for clogging seems not to change in increase of holding time. Similarly, critical solid fraction for clogging is in the range from 20% to 45%. Therefore, clogging seems to be occurring in this region of solid fraction.

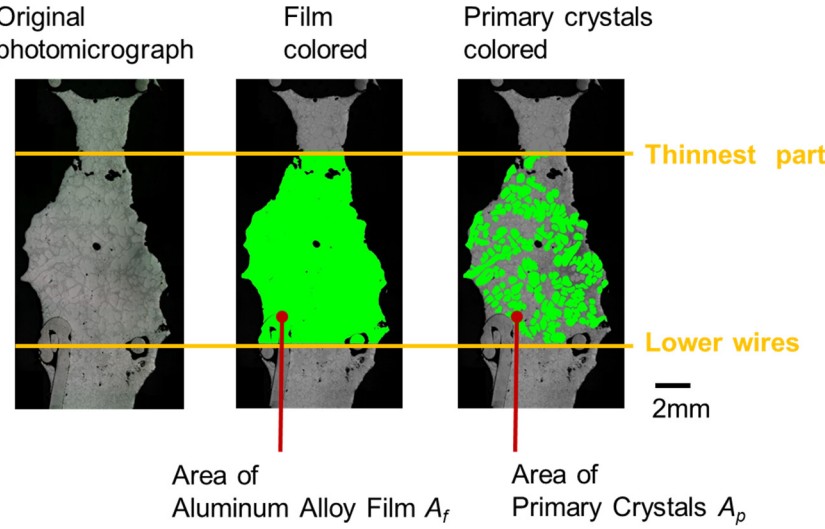

**Figure 11.** Evaluations of area of aluminum alloy film and primary crystals.

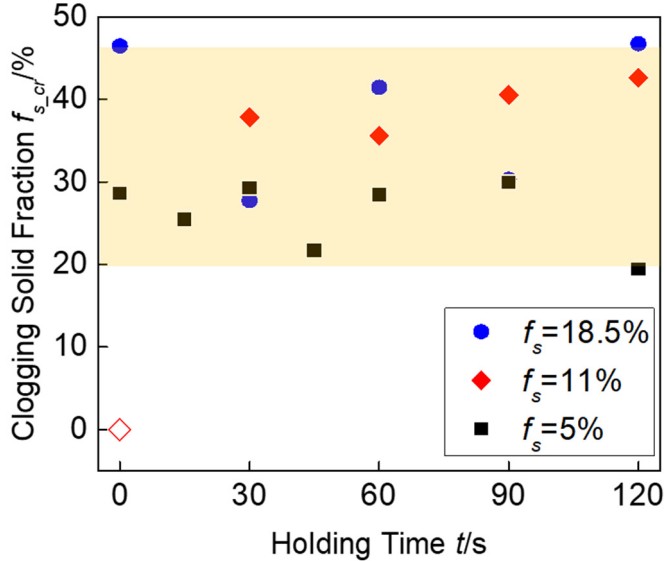

**Figure 12.** Relationship between critical solid fraction for clogging $f_{s\_cr}$ and holding time for initial solid fraction $f_s$ of 5%, 11%, and 18.5%.

Clogging of thickener has also been reported for the Melt route [13]. However, bridge formation of thickener attributed to oxide film is thought to be the main stabilization mechanism in Melt route [12]. For Semi-solid route, since the size of thickener is large, clogging is possible to occur below the thinnest part of film and it is assumed that film is not required to be thin for the clogging to occur.

### 4.3. Fluid Flow and Clogging of Primary Crystals

According to the results in Figure 5 clogging occurs at a certain timing, which becomes earlier with increasing solid fraction. For solid fraction of 18.5%, clogging seems to have occurred right after the pull-up. However, for the lower solid fractions, drainage following the model formula occurs in a pulled-up film until the clogging. Figure 13 which is a qualitative representation based on Figure 5 summarizes the schematic and graph of this phenomenon. The solid lines show the actual change of normalized cross-sectional area $A/A_0$. The broken lines and curved solid lines show the values obtained from model formula.

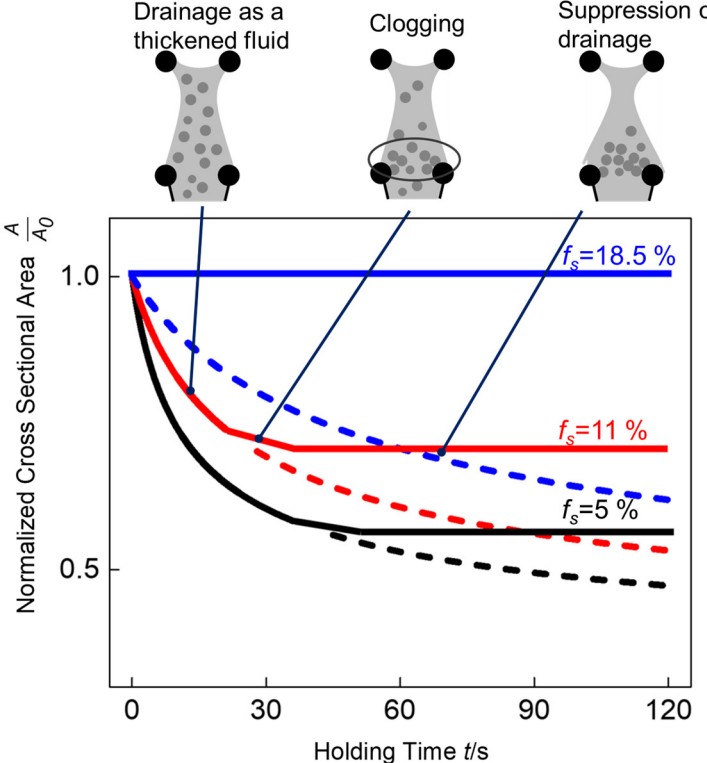

**Figure 13.** Clogging time in model formula and schematic of clogging mechanism. The solid lines show the actual change of normalized cross-sectional area $A/A_0$. The broken lines and curved solid lines show the values obtained from model formula.

## 5. Conclusions

The stabilization mechanism of cell wall in Semi-solid route was evaluated using aluminum alloy films. The results can be summarized as follows.

1.  Pulled-up film drainages as a thickened melt. Since this could be explained as Poiseuille flow, drainage rate becomes slower with increasing solid fraction.
2.  At a certain timing clogging occurs and drainage is largely suppressed. Clogging is occurring in the range of solid fraction of 20–45% under the thinnest part of the film.
3.  The clogging occurs earlier as the solid fraction increases. For solid fraction $f_s$ = 18.5% clogging seems to be already occurring in holding time $t$ = 0 s.

**Author Contributions:** Conceptualization, T.K. and S.S.; methodology, T.K., A.K., T.O. and S.S.; validation, T.K., A.K., T.O. and S.S.; formal analysis, T.K.; investigation, T.K., A.K., T.O. and S.S.; data curation, T.K. and S.S; writing—original draft preparation, T.K.; writing—review and editing, T.K., S.T. and S.S.; visualization, T.K.; supervision, S.S.; project administration, S.S.; funding acquisition, T.K. and S.S. All authors have read and agreed to the published version of the manuscript.

**Funding:** This study was supported by the Grant-in-Aid the Light Metal Educational Foundation.

**Acknowledgments:** The authors thank The Light Metal Educational Foundation for suppling the pure Aluminum ingots used in this study.

**Conflicts of Interest:** The authors declare no conflict of interest.

## Appendix A

Here, derivation of geometry model is expressed. As expressed in Section 4.1.1 and Figure 8, a pulled-up film is approximated as a rectangle. Therefore, the cross-sectional area of the film at this point can be expressed as Equation (A1).

$$A_0 = hx_0 \tag{A1}$$

Here, $A_0$ is the initial cross-sectional area, $h$ is height of rectangle, and $x_0$ is width of rectangle. Then, the side wall of the film will start to become a curved surface as the drainage progresses. At this point cross section of the film can be shown as Figure A1. Here, $\theta$ is the curvature radius of the film, $x_t$ is the width of middle point of rectangle which would change continuously.

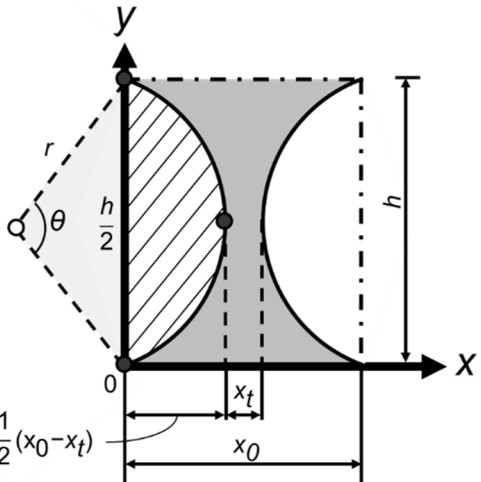

**Figure A1.** Coordinate axes of cross section of aluminum alloy film during progress of drainage.

Figure A1 also shows the coordinate axes of aluminum alloy film. Rectangle inside chain line shows the initial cross section. To obtain the cross-sectional area $A$ as a formula, area of shaded part $S$ in Figure A1 is obtained as follows. The equation of a circle with this curvature can be shown as Equation (A2) since the circle is known to pass through three dark points. Therefore, the radius of the circle can be shown as Equation (A3). Middle point of circle is expressed as light point in Figure A1.

$$\left\{ x - \frac{(x_0 - x_t)^2 - h^2}{4(x_0 - x_t)} \right\}^2 + \left( y - \frac{h}{2} \right)^2 = \frac{1}{16(x_0 - x_t)^2} \left\{ (x_0 - x_t)^2 + h^2 \right\}^2 \tag{A2}$$

$$r = \frac{1}{4(x_0 - x_t)} \left\{ (x_0 - x_t)^2 + h^2 \right\} \tag{A3}$$

Then the area of the light gray triangle $S'$ in Figure A1 can be obtained as Equation (A4) from Heron's formula using three sides of triangle.

$$S' = \frac{h}{8(x_0 - x_t)} \left\{ h^2 - (x_0 - x_t)^2 \right\} \tag{A4}$$

Since the area of triangle can also be shown as Equation (A5), the curvature radius $\theta$ can be obtained as Equation (A6) from Equations (A4) and (A5).

$$S' = \frac{1}{2}r^2 \sin\theta \tag{A5}$$

$$\theta = \operatorname{Arcsin} \frac{4h(x_0 - x_t)\left\{h^2 - (x_0 - x_t)^2\right\}}{\left\{h^2 + (x_0 - x_t)^2\right\}^2} \tag{A6}$$

From the curvature radius, the area of the sector composed of the shaded part and the light gray triangle can be obtained as Equation (A7). Then, by subtracting Equation (A4) from Equation (A7), the area of the shaded part can be obtained as Equation (A8). In the equations, $S$ is the shaded part which shows the area of outer part the of curved film.

$$(S + S') = \frac{\pi}{16(x_0 - x_t)^2}\left\{h^2 + (x_0 - x_t)^2\right\}^2 \frac{\theta}{360°} \tag{A7}$$

$$S = \frac{1}{8(x_0 - x_t)}\left[\frac{\pi}{2(x_0 - x_t)}\left\{h^2 + (x_0 - x_t)^2\right\}^2 \frac{\theta}{360°} - \left\{h^2 - (x_0 - x_t)^2\right\}h\right] \tag{A8}$$

Since there are curved parts at both side of film, the total area of the outer part of the curved film can be shown as Equation (A9).

$$2S = \frac{1}{4(x_0 - x_t)}\left[\frac{\pi}{2(x_0 - x_t)}\left\{h^2 + (x_0 - x_t)^2\right\}^2 \frac{\theta}{360°} - \left\{h^2 - (x_0 - x_t)^2\right\}h\right] \tag{A9}$$

Finally, by subtracting Equation (A9) from Equation (A1), the cross-sectional area of the film with a curvature can be shown as Equation (A10) with $\Delta x$ defined as Equation (A11).

$$A = A_0 - 2S = hx_0 - \frac{1}{4\Delta x}\left\{\frac{\pi}{2\Delta x}\left(h^2 + \Delta x^2\right)^2 \frac{\theta}{360°} - \left(h^2 + \Delta x^2\right)h\right\} \tag{A10}$$

$$\Delta x = x_0 - x_t \tag{A11}$$

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
