# Peer review of "Stabilization Mechanism of Semi-Solid Film Simulating the Cell Wall during Fabrication of Aluminum Foam"

_metals, doi:10.3390/met10030333_

Round 1
Reviewer 1 Report
The manuscript entitled: Stabilization Mechanism of Semi-solid Film Simulating the Cell Wall during Fabrication of Aluminum Foam aims to elucidate the stabilization mechanism of cell wall od semi-solid route using aluminum alloy film. The analysis is interesting, however, it lacks scientific depth. The mechanisms proposed are not convincing. In addition, it may be apt, if the authors compare the stabilization mechanisms in the foams produced by semi-solid route with the melt route and highlight the differences, so that the importance of the presence of solid material in the semi-solid route may be appreciated.
Author Response
February 23rd, 2020
Dear Reviewer
Thank you for consideration for our Article. Also, we are sorry for the late submission. The changes made in the manuscript is as follows.
The correction for each comment was listed in the answer sheet.
Changes we made from reviewer’s comment are obvious in the manuscript submitted as supplementary file because we colored the changed part in red.
Again, thank you for your consideration of our manuscript. We look forward to hearing from you.
Sincerely,
Takashi KUWAHARA (Corresponding author)
Graduate School of Fundamental Science and Engineering, Waseda University,
59-305, 3-4-1 Okubo, Shinjuku, Tokyo 169-8555, Japan
Email: takuwahara@ruri.waseda.jp
TEL: +81-3-5286-8126

Reviewer 2 Report
This article presents experimental results on the draining of a semi-solid Al film through a device, designed to simulate the drainage around the pores of a foam during solidification. The geometry change of the film during drainage is measured and related to the mechanism of clogging, through microscopic observation of solidified crystals, and geometrical calculations. Overall the article presents interesting and useful results to understand the phenomenon of draining during foam solidification, thus I recommend that it is accepted for publication, subject to the below edits:
English language editing is needed throughout. Mention clearly that the design of the ring-wire device originates from ref [13], and the authors are adopting it to study the semi-solid densification phenomena. Can the authors comment on how close the observations of the drainage through this device correspond to those in a 3D channel? Line 153: “...solid fraction...” Line 183: Why was L assumed constant? Can the validity of this assumption be verified through measurements of the film’s cross-section? Eq 4. What values of viscosity and density were used? How were they obtained? The quantification of clogging is very unclear. Can the authors clarify the following: How are the clogging solid fraction and the critical solid fraction fs_cr defined, and how are they different? How was fig 13 obtained from Fig 5? Is fig 13 a qualitative representation of fig 5?Author Response
February 23rd, 2020
Dear Reviewer
Thank you for consideration for our Article. The changes made in the manuscript is as follows.
The correction for each comment was listed in the answer sheet.
Changes we made from reviewer’s comment are obvious in the manuscript submitted as supplementary file because we colored the changed part in red.
Again, thank you for your consideration of our manuscript. We look forward to hearing from you.
Sincerely,
Takashi KUWAHARA (Corresponding author)
Graduate School of Fundamental Science and Engineering, Waseda University,
59-305, 3-4-1 Okubo, Shinjuku, Tokyo 169-8555, Japan
Email: takuwahara@ruri.waseda.jp
TEL: +81-3-5286-8126

Round 2
Reviewer 1 Report
The authors have revised the manuscript in a satisfactory manner and it may now be accepted for publication.